# Size control in mammalian cells involves modulation of both growth rate and cell cycle duration

Clotilde Cadart [1,2], Sylvain Monnier [1,3], Jacopo Grilli[4,5], Pablo J. Sáez [1,2], Nishit Srivastava[1,2], Rafaele Attia[1,2], Emmanuel Terriac[1,11], Buzz Baum[6,7], Marco Cosentino-Lagomarsino [8,9,10] & Matthieu Piel [1,2]

Despite decades of research, how mammalian cell size is controlled remains unclear because of the difficulty of directly measuring growth at the single-cell level. Here we report direct measurements of single-cell volumes over entire cell cycles on various mammalian cell lines and primary human cells. We find that, in a majority of cell types, the volume added across the cell cycle shows little or no correlation to cell birth size, a homeostatic behavior called "adder". This behavior involves modulation of G1 or S-G2 duration and modulation of growth rate. The precise combination of these mechanisms depends on the cell type and the growth condition. We have developed a mathematical framework to compare size homeostasis in datasets ranging from bacteria to mammalian cells. This reveals that a near-adder behavior is the most common type of size control and highlights the importance of growth rate modulation to size control in mammalian cells.

[1] Institut Curie, PSL Research University, CNRS, UMR 144, F-75005 Paris, France. [2] Institut Pierre-Gilles de Gennes, PSL Research University, F-75005 Paris, France. [3] Univ. Lyon, Université Claude Bernard Lyon 1, CNRS, Institut Lumière Matière, F-69622 Villeurbanne, France. [4] Department of Ecology and Evolution, University of Chicago, 1101 E 57th Street, Chicago, IL 60637, USA. [5] Santa Fe Institute, 1399 Hyde Park Road, Santa Fe, NM 87501, USA. [6] MRC Laboratory for Molecular Cell Biology, UCL, London WC1E 6BT, UK. [7] Institute of Physics of Living Systems, UCL, London WC1E 6BT, UK. [8] Sorbonne Universités, Université Pierre et Marie Curie, Paris F-75005, France. [9] CNRS, UMR 7238 Computational and Quantitative Biology, Paris F-75005, France. [10] FIRC Institute of Molecular Oncology (IFOM), Milan 20139, Italy. [11] Present address: INM-Leibniz Institute for New Materials, Campus D2 2, 66123 Saarbrücken, Germany. Correspondence and requests for materials should be addressed to M.C.-L. (email: marco.cosentino-lagomarsino@upmc.fr) or to M.P. (email: matthieu.piel@curie.fr)

There is little consensus about the way mammalian cells control their size[1,2]. Studies of single-celled yeast and bacteria have revealed that in order to achieve size homeostasis, cells must modulate the amount of growth produced during the cell cycle such that, on average, large cells at birth grow less than small ones. Size homeostasis can be exemplified by three simple limit cases: the sizer, the adder and the timer. Perfect size control has been reported for the fission yeast, S. Pombe[3], where a size threshold (sizer) was proposed to control the entry into mitosis[4]. By contrast, an "adder" mechanism relies on the addition of a constant volume at each cell cycle that is independent of initial size[5,6], causing cells to converge on an average size after a few generations. This behavior has been reported for several types of bacteria, cyanobacteria and in budding yeast[7–11]. Finally, if cells grow exponentially for a constant amount of time (a "timer" mechanism), large cells grow more than smaller ones and sizes diverge rapidly. Alternatively, if cells grow linearly, a timer results in cells growing by the same amount each cell cycle, therefore maintaining size homeostasis[12].

In bacteria and yeast, the development of high-throughput single live cell imaging has provided a wealth of measurement which, together with the development of theoretical models enabled great progress in the characterization of size control in these organisms[11,13–20]. Similar progress has yet to be made in mammalian cells which have complex and fluctuating shapes. To date, most studies on mammalian cells have relied on population level measurements[12,21–24]. These include attempts to extrapolate growth dynamics from size measurements at fixed time points across a population[24–27]. Recently, a variety of parameters such as cell dry mass[26,28,29], buoyant cell mass[30] and cell density[31], have been used as proxies for size at the single-cell level, mostly through indirect techniques. Among these recent studies are measurements of single-cell size at specific times in the cell cycle[32] or through complete cell cycles[28–30]. Although most data in unicellular organisms were obtained on cell volume, and most size-sensing mechanisms currently debated are thought to involve concentration-dependent processes[19,33–35], measurements of volume trajectories on single cycling mammalian cells have not been reported yet and it is thus unclear whether volume and mass are similarly relevant for size control. Moreover, the paucity of direct and dynamic measurements on single live cells has limited the identification of regulatory processes leading to size control in mammalian cells.

Similarly to unicellular organisms, mammalian cells have been hypothesized to control their size via a modulation of cell cycle duration. Specifically, an adaptation of G1 duration as a function of cell size has been proposed by a series of indirect[21,23,25,37] and one direct[32] work. Other studies on mammalian cells have reported negligible changes in cell cycle timing and have hypothesized that changes in growth speed may contribute to cell size control[24,27] (we define here growth speed as the evolution of size as a function of time, and growth rate as the evolution of growth speed as a function of size). Direct observation of a convergence of growth speed at the G1/S transition was seen in lympho-blastoid cells[30] but how this leads to an effective cell size homeostatic behavior was not characterized. The idea that growth speed modulations could play a role in mammalian cells size control was not tested directly and its contribution to overall size homeostasis has not been compared to that of time modulation. Moreover, the contribution of S-G2 duration in size control and the effective homeostatic behavior from birth to mitosis has not been characterized yet.

To address these questions as directly as possible, we recently developed two methods to precisely measure the volume of large numbers of single live cells over several days[37–39]. In this study, we used these tools to track single-cell volume growth over complete cell cycles. We characterize the homeostatic behavior of a variety of cultured and primary mammalian cells and show that they behave like adders (or near adders). We then quantify the modulation of time (in G1 and S-G2) and growth rate that contribute to size control. Finally, we develop a quantitative framework that characterizes the relative contributions of timing and growth modulation to size homeostasis from bacteria to mammalian cells.

## Results

**Single-cell volume measurement over entire cell division cycles.** The homeostatic behavior of cells is identified by assessing the relation, for single cells, between their size at mitotic entry and their size at birth. This relation has never been reported for freely growing mammalian cells in culture.

To establish this relation, it is necessary to track single proliferating cells and measure the volume of the same cell at birth and at mitotic entry. We implemented two distinct methods to obtain these measures. First, we grew cells inside micro-channels of a well-defined cross-sectional area (Supplementary Fig. 1a, and ref.[37].), as was recently reported for immune cells[32]. In such a geometry, dividing cells occupied the whole section of the channels and had a cylinder shape, thus we could infer their volume from their length. The second method we used is a Fluorescence eXclusion measurement method (FXm) to measure volume[38,39] (Fig. 1a, Supplementary Movie 1). In this technique, cells are seeded in a chamber of known height and a fluorescent probe that does not enter the cell is added to the culture media. The fluorescence intensity is negatively proportional to the height of the cell and the exact volume of the cell can therefore be calculated (Fig. 1a). In previous work, we validated the FXm method and showed that it allows single-cell volume measurement, independently of cell shape[38,39]. Here, we optimized the method for long term recording and automated analysis of populations of growing cells (controls are presented in Supplementary Fig. 1b and Method). This method has several advantages (reviewed in ref.[40]): compared with microchannels, it does not require growing cells in a very confined environment, which is thought to constrain growth to a linear pattern[32], and it is more precise. It also produces complete growth trajectories for single cells (Fig. 1b, c and Supplementary Fig. 1c). Visual inspection of the movies was used to determine key points in the cell division cycle for each single-cell tracked. Volume at birth was defined as the volume of a daughter cell 40 min after cytokinesis onset, while volume at mitotic entry was defined as volume of the same cell 60 min prior to the next cytokinesis onset (Fig. 1b, Supplementary Fig.1d,e). Analysis of growth speed as a function of size, for a large number of single cells and cell aggregates showed that the average growth speed increased linearly with cell size (Supplementary Fig. 1f). This supports that on average cells grew faster than linearly and is compatible with a (mean) exponential mode of growth, as previously reported in some cases for freely growing cells[26,27,29,30] (note that other modes of growth that are super-linear, may also describe our data, as explained in Supplementary Note 1, but for simplicity we approximate to exponential growth).

We studied two types of cancerous epithelial cell lines (HT29 wild-type (HT29-wt) and HT29 expressing hgeminin-mcherry (HT29-hgem), HeLa expressing hgeminin-GFP (HeLa-hgem) and HeLa expressing MyrPalm-GFP H2B-mcherry (HeLa-MP)), one B lymphoblast cancerous cell line (Raji), one non-cancerous aneuploid epithelial cell line (MDCK expressing MyrPalm-GFP (MDCK-MP)), and one hTERT-immortalized epithelial cell line (RPE1). For each experiment performed, the dataset was checked for quality: we verified that the distribution of volumes at birth

and the average growth speed did not change throughout the experiment, and that these values did not change from one experiment to another (Fig. 1d and Supplementary Fig. 1g). Note that we kept one dataset which showed a significant, but small, decrease in volume through the course of the experiment, because despite optimization, we could not avoid some internalization of dextran by these cells (Supplementary Fig. 1g, HeLa-hgem cells, Supplementary Movie 1). This decrease was however below 10% at the end of experiments lasting 40 h, and thus could not impact our analysis. We were able, with these methods, to produce fully validated high-quality datasets of single-cell volume over entire cycles, which can be further used to ask elementary questions on volume homeostasis for proliferating cultured mammalian cells.

**A near-adder behavior is observed in mammalian cells**. The effective homeostatic behavior can be assessed phenomenologically by quantifying the relation between added volume during the cell cycle and volume at birth (Fig. 2a). If cells double their volume (i.e., in the case of exponentially growing cells with a timer), the added volume is equal to the volume at birth, thus the two values linearly correlate with a slope of 1, and the final vs. initial volume plot shows a slope of 2. On the other hand, if cells are perfectly correcting for differences in size (sizer), the added volume is smaller for larger cells, and the slope of this plot is negative, while the final volume is identical for all cells independently of their initial volume.

The six cell types we analyzed (HT29, HeLa, MDCK, Raji, RPE1 and L1210), behaved neither as timers nor as sizers

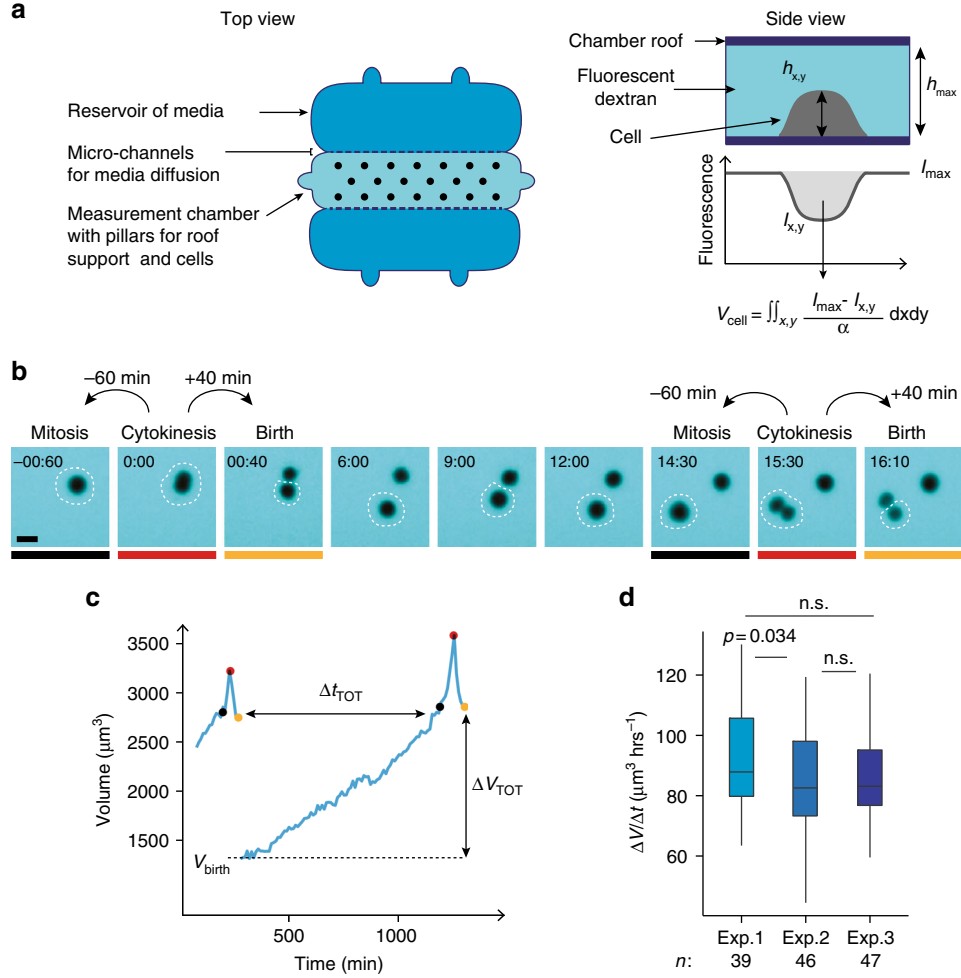

**Fig. 1** Single-cell volume tracking over entire cell division cycles. **a** Principle of the fluorescence exclusion volume measurement method (FXm). Left: top view of the measurement chamber used for 50 h long time-lapse acquisitions (see Methods). Right: side view of the chamber and principle of the measurement. Fluorescence intensity at a point $I_{x,y}$ of the cell is proportional to the height of the chamber minus the height $h_{x,y}$ of the cell at this point. Fluorescence intensity $I_{max}$ is the intensity under the known height of the chamber roof $h_{max}$, where no object excludes the fluorescence. Integration of fluorescence intensity over the cell area gives the cell volume $V_{cell}$ after calibrating the fluorescence intensity signal $\alpha = (I_{max} - I_{min})/h_{max}$ (see Methods). **b** Sequential images of a HT29-wt cell acquired for FXm. Mitosis and birth are defined as the time points 60 min before and 40 min after cytokinesis respectively (see Methods). The white dashed circle indicates the cell measured in Fig. 1c, the colored lines indicate the time points highlighted by circles of the same color in Fig. 1c. Time is in hours:minutes. Scale bar is 20 μm. **c** Single HT29-wt cell growth trajectory (volume as a function of time) and key measurement points (see Methods). The time points shown in Fig. 1b and underlined in gray, red, or yellow are indicated by points of matching colors on the curve: the gray points correspond to volume at mitotic entry, the red points correspond to volume at cytokinesis and the yellow points to volume at birth. $\Delta t_{TOT}$ is the total duration of the cell division cycle from birth to mitosis and $\Delta t_{TOT}$ is the total added volume. **d** Average growth speed for three independent experiments with HT29-wt cells. $n = 39$ (exp. 1), $n = 46$ (exp. 2), $n = 47$ (exp. 3). The p-values are the result of a pairwise t test comparing the means. See also Supplementary Figure 1 and Supplementary Movie 1

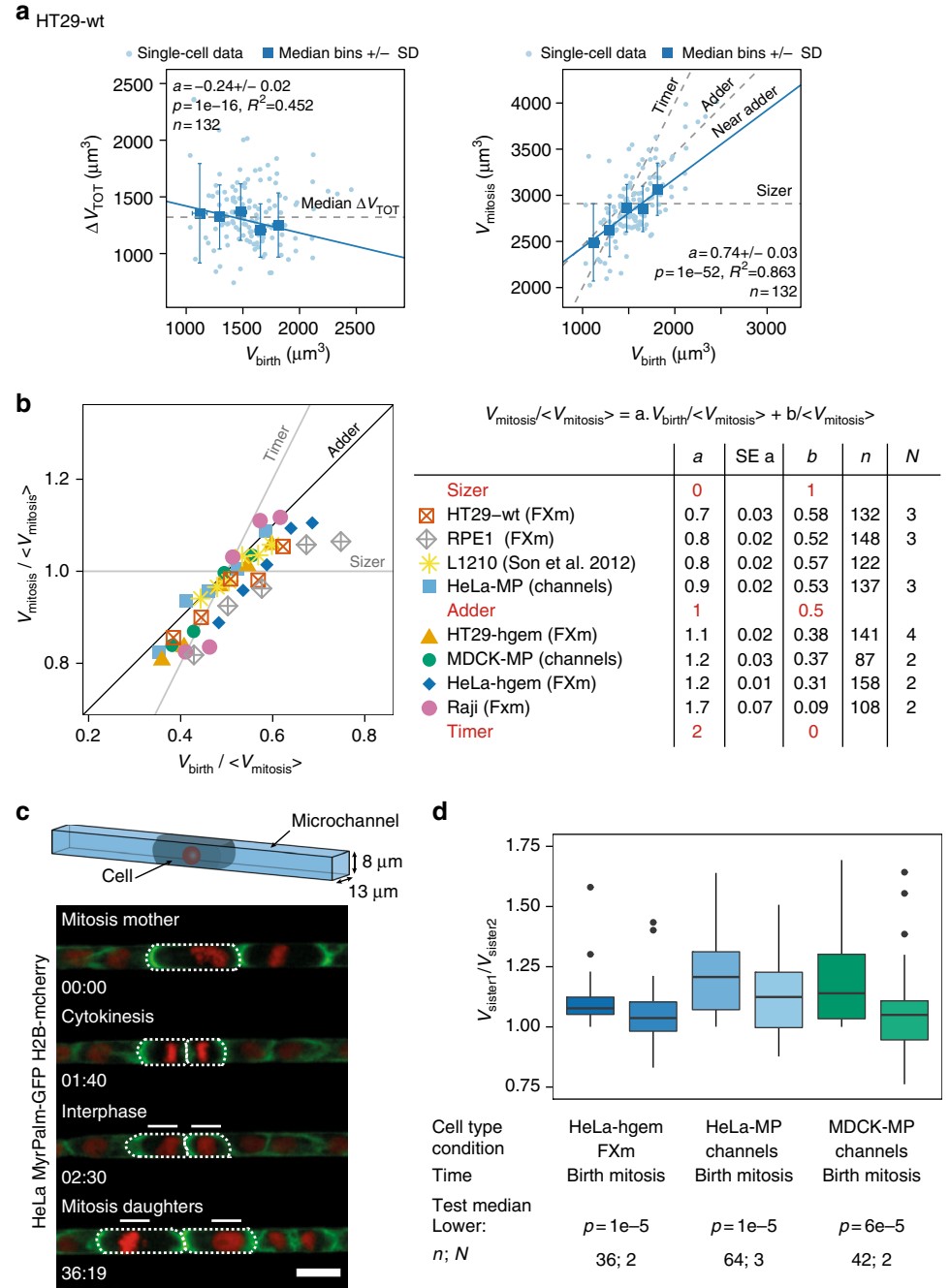

**Fig. 2** Adder or near-adder behavior in cultured mammalian cells. **a** Left: total volume gained during one cell division cycle $\Delta t_{TOT}$ vs. volume at birth $V_{birth}$ for wild-type HT29 cells ($N = 3$). Right: volume at mitosis $V_{mitosis}$ vs. $V_{birth}$. Dashed gray lines show the expected trends in case of a sizer, an adder, and a timer. Blue lines: linear fit on the binned data weighted by the number of observations in each bin. **b** Left graph: plot of volume at mitosis vs. volume at birth rescaled by the mean volume at mitosis for various cultured mammalian cell lines. Ideal slopes for stereotypical homeostatic behaviors are shown as black and gray lines. The points are median bins (see Supplementary Fig. 2b for equivalent graphs with single points). For each cell type, a linear fit $V_{mitosis} = aV_{birth} + b$ is made on the bins weighted by the number of observation in each bin. Right table: estimates from the linear regression for each cell type: $a$ (slope coefficient), s.e. $a$ (standard error for $a$), $b$ (slope intercept). The theoretical slope coefficients and intercepts expected in case of sizer, adder, or timer are also indicated. L1210 are mouse lymphoblastoid cells from ref.[33]. Apart from the L1210 cells buoyant mass, data are volumes acquired with either the FXm or the microchannel methods. **c** Top: scheme of a cell confined in a microchannel (nucleus in red). Bottom: sequential images of an asymmetrically dividing HeLa cells expressing MyrPalm-GFP (plasma membrane, green) and Histon2B-mcherry (nucleus, red) growing inside a microchannel. The outlines of the cell of interest and its daughters are shown with white dotted lines. Daughter cells are indicated with solid white bars. Scale bar is 20 μm. Time is hours:minutes. **d** Ratio of volume in pairs of sister cells at birth and mitosis for MDCK-MP and HeLa-MP cells growing inside microchannels. Control, in non-confined condition, corresponds to HeLa-hgem cells measured with FXm. A Wilcoxon signed rank test was performed to test that the median ratio was lower from birth to mitosis in each condition. See also Supplementary Figure 2 and Supplementary Movie 2

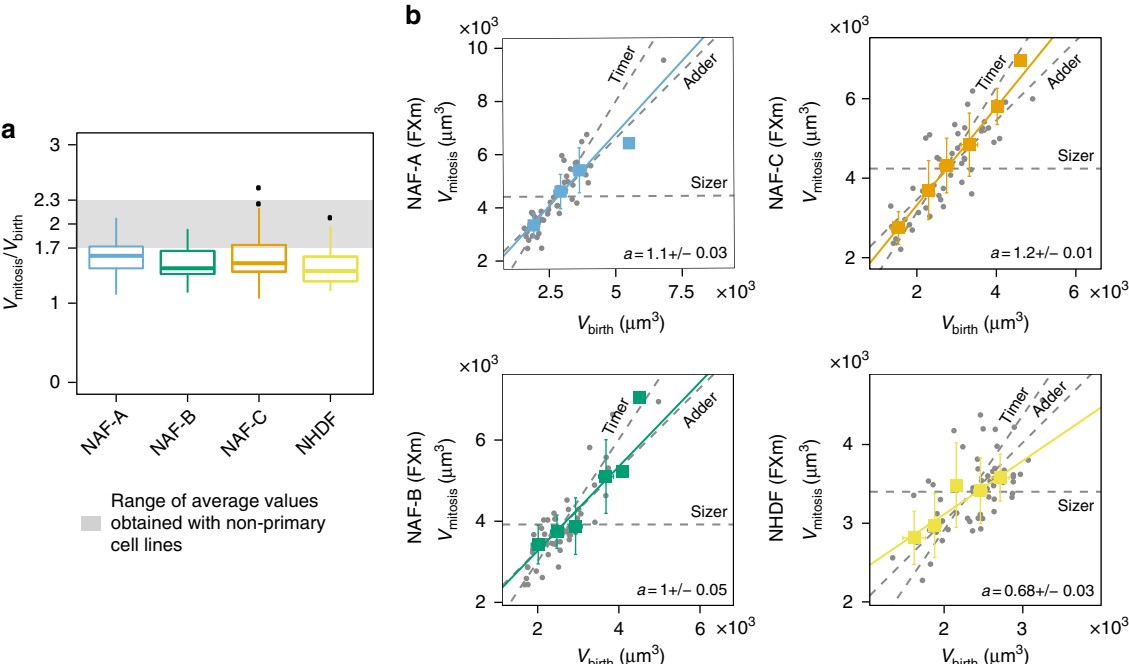

**Fig. 3** Near-adder behavior in primary human cells. **a** Boxplot showing the distribution of over replicative growth (volume at mitosis divided by volume at birth) for three samples of NAF and NHDF primary cells. NAF-A:, $n = 48$, $N = 2$; NAF-B: $n = 53$, $N = 2$; NAF-C: $n = 53$, $N = 2$; NHDF: $n = 56$, $N = 3$. **b** Volume at mitosis as a function of volume at birth for three samples of NAF and NHDF primary cells. Dashed lines are visual guides for the timer timer (assuming exponential growth, slope $= \langle V_{mitosis}/G_{1/S} \rangle$, intercept $= 0$), adder (slope $= 1$, intercept $= \langle \Delta V_{S-G2} \rangle$) and sizer (slope $= 0$, intercept $= \langle V_{mitosis} \rangle$). Solid lines represent linear fits on the bins (colored squares) weighted by the number of observations in each bin

(Supplementary Fig. 2a-c). With the exception of Raji cells, which showed a large dispersion of added volumes, and for which added volume correlated positively with volume at birth (Supplementary Fig. 2a), we instead found that added volume showed no correlation (HT29-hgem, HeLa-hgem, HeLa-MP, and MDCK-MP) or a weak negative correlation (HT29-wt and RPE1) with volume at birth (Fig. 2a, Supplementary Fig. 2a). Consistently, the volume at mitotic entry was linearly correlated to volume at birth, with a slope ranging from 0.7 to 1.2 (Fig. 2b, Supplementary Fig. 2b). This observation was also reproduced when analyzing previously published results obtained on lymphoblastoid L1210 cells (kindly shared by the authors[30]). (Note that in the rescaled plot shown in Fig. 2b, RPE1 and HeLa-hgem do not overlap with the other datasets because they displayed a lower overall doubling ratio $V_{mitosis}/V_{birth}$ (discussed in Methods and Supplementary Fig. 2d)). Thus, with the exception of Raji cells, five of the six cell lines studied here displayed an adder or near-adder type of homeostatic behavior, reminiscent of what was already described for several bacterial species and for the buds of budding yeast cells[7,8,11].

In bacteria, this weak form of volume homeostasis was shown to compensate for asymmetries in sizes occurring at division[5,7]. A direct prediction is that, after an asymmetric division, the difference in size of the two daughter cells would be reduced by half in the following cycle, but not fully corrected. To confirm the observation of the near-adder in cells with large asymmetries in size, we artificially induced asymmetric divisions by growing two different cell types (HeLa and MDCK) inside microchannels (Supplementary Fig. 1a, Supplementary Movie 2). Confinement prevents mitotic rounding, which leads to errors in the mitotic spindle positioning and ultimately generates uneven division of the mother cell (Fig. 2c, d, refs.[37,41]). We then compared the asymmetry in volume, at birth and at the next mitosis, between pairs of daughter cells. For both cell types, the level of volume asymmetry at birth was higher in channels than in cells that

divided outside of the channels, and was significantly reduced at entry into the next mitosis, but not completely compensated for (Fig. 2d), as predicted by a near-adder behavior. In conclusion, this first analysis revealed that most cultured mammalian cell lines display a near-adder behavior.

**Primary human cells behave as near-adder**. We then wondered whether the observation of the near-adder extended to primary cells and repeated our experiments on normal associated fibroblasts (NAFs) and normal human epidermal fibroblasts (NHDFs). These cells come from healthy tissues in patients, and present the advantage of not being mutated for any growth or cell cycle pathways. However, they are a complex experimental system because they are very heterogeneous and out of steady state in culture where they progressively stop dividing. As expected, NAF and NHDF were highly variable both in cell cycle duration and volume distribution (Supplementary Fig. 3a-c). In the FXm chambers, they also showed a low overall doubling ratio $\langle V_{mitosis}/V_{birth} \rangle$ that ranged from 1.5 to 1.6 (Fig. 3a), indicating that they were not at steady state (see Methods and Supplementary Fig. 3d-f). It however remained possible to characterize their homeostatic behavior. The analysis of the relationship between volume at mitosis and volume at birth revealed that NHDF and three different samples of NAF, similar to immortalized cell lines, all behaved as adder or near-adder (Fig. 3b, Supplementary Fig. 3g-j).

In conclusion, the adder or near-adder is the most common homeostatic behavior observed in a variety of immortalized and primary mammalian cells. Importantly, a near-adder observed at the phenomenological level does not necessarily imply the existence of a molecular mechanism "counting" added volume. The most recent findings in unicellular organisms instead suggest that the near-adder may emerge from the combination of several mechanisms acting in parallel or sequentially during the cell cycle[42].

**Modulation of G1 duration contributes to size control**. Modulations of cell cycle duration as a function of size are the basis of size regulation in unicellular organisms. In animal cells, similarly to budding yeast[33,43–46], indirect population level approaches suggested that modulation of G1 duration is important for size control[21,23,36] and that this occurs through the p38-MAPK pathway[36]. Recent direct measurements confirmed this hypothesis, using confinement inside microchannels[32], a system that caused cells to grow linearly and that did not allow the study of homeostatic behavior in S-G2 or over the whole-cell cycle. Hence these points have yet to be investigated for cells that grow in regular culture conditions, such as the FXm chambers where cells grew exponentially (Supplementary Fig. 1f).

To investigate the contribution of modulations of G1 and S/G2 phase duration in size control, we combined cell volume measurements on HT29 and HeLa cells with a classical marker of cell cycle phases, hgeminin, which accumulates in the cell nucleus at S-phase entry[47] (Fig. 4a, Supplementary Fig. 4a and Supplementary Movie 3). HeLa expressing hgeminin-mcherry (HeLa-hgem) on average cycled faster than HT29 expressing hgeminin-mcherry (HT29-hgem) (Fig. 4b). This difference was mostly the consequence of a longer and more variable G1 phase in HT29-hgem (HT29-hgem, CV = 53%, HeLa-hgem, CV = 18%) while S-G2 duration showed little variation for both cell types (HT29-hgem, CV = 18%, HeLa-hgem, CV = 17%) (Fig. 4b, Supplementary Fig. 4b).

Despite this quantitative difference in average duration of G1, HT29-hgem and HeLa-hgem displayed common traits qualitatively. For both cell types, G1 duration and added volume in G1 correlated negatively with cell volume at birth (Fig. 4c–f), indicative of the existence of size control via a modulation of G1 duration. Consistently, the volume at the end of G1 plotted against volume at birth showed a slope below 1 (HT29-hgem: $a = 0.71 \pm 0.01$, HeLa-hgem: $a = 0.69 \pm 0.01$, slope ± standard error ), suggesting an intermediate strength of size control, between the adder and the sizer (Fig. 4g, h).

This analysis also suggests that there is a minimal duration of the G1 phase, an observation that reproduces recent results in microchannels[32]. Indeed, for HT29-hgem, smaller cells showed a wider dispersion of G1 duration while larger cells tended to spend only a minimal time in G1 (about 4 h) (Fig. 4c). This is well illustrated by the cumulative distribution functions of the time spent in G1 for three ranges of volumes at birth (Fig. 4i). HeLa-hgem which on average cycle faster, seemed, by comparison with the HT29-hgem cells, to all cycle very close to a similar minimum G1 duration (about 4 h) (Fig. 4d–j).

Together, these results provide evidence for size control of intermediate strength between the adder and the sizer in G1 that involves a modulation of G1 duration. Additionally, modulation of G1 timing appears limited by the existence of a minimum G1 duration.

**Modulation of S-G2 duration in HeLa but not HT29**. In order to test the existence of size control in S-G2, we repeated the same analysis as done in G1. For HT29-hgem cells, S-G2 duration was not correlated with volume at G1/S (Fig. 5a) and showed little cell-to-cell variation (Fig. 4b). This is typically indicative of a "timer" behavior. As expected from the combination of a timer and exponential growth, we found a positive correlation between added volume in S-G2 and volume at the G1/S transition (Fig. 5b) and the slope of volume at mitosis vs. volume at G1/S was very close to the expected slope for a timer (Supplementary Fig. 5a). HeLa-hgem cells showed a different behavior. For these cells, S-G2 duration was negatively correlated with volume at the G1/S transition (Fig. 5c) and added volume in S-G2 was not correlated

with volume at G1/S (Fig. 5d). Hence, these cells displayed a near-adder behavior in S-G2, as confirmed by the plot of volume at mitosis vs. volume at G1/S (Supplementary Fig. 5b).

Our observation of some control on size in S-G2 in HeLa cells cannot be compared with previous results, which focused only on size control in G1[21,23,32]. Following the strategy proposed by Chandler-Brown and coworkers[45], we tested the hypothesis of a "mechanistic-adder", i.e., that the rate-limiting process for cell-cycle completion is the addition of a nearly constant volume from birth to mitosis. Since in this hypothesis added volume in S-G2 should perfectly match added volume in G1, so that $\Delta V_{G1} + \Delta V_{S-G2} = \Delta V_{tot} = $ Constant, one can test the relation of $\Delta V_{S-G1}$ and $\Delta V_{G1}$, and a slope of $-1$ would correspond to the mechanistic adder[45]. Contrary to budding yeast, for both HT29-hgem and HeLa-hgem (Fig. 5e, f), the slope was generally negative and followed a trend that might be compatible with the mechanistic adder prediction, except for a few strong outliers in HT29-hgem cells.

Thus, our analysis of growth in S-G2 revealed an unsuspected role of modulation of S-G2 duration for size control in HeLa cells, while S-G2 was closer to a timer in HT29 cells. Whether this additional size control mechanism is cell-type dependent or rather specific to faster-growing cells will require further investigation. Taken together with the analysis of G1 phase (Fig. 4), these results show that modulation of G1 and/or S-G2 duration contributes to size control in cells that on average grow exponentially but that the two cell types we studied rely differently on these mechanisms in order to achieve a similar "near-adder" effective behavior (Fig. 5g, h).

**Large cells do not adapt G1 duration**. Figure 4 shows a lower limit on the duration of G1 phase for the largest HT29-hgem cells (Fig. 4c–i) and fast cycling HeLa-hgem cells (Fig. 4d–j), which implies that, if growth was exponential and homeostasis mechanism limited to modulations of time, it would not be possible to have homeostasis in G1 for larger cells. To further test this, we produced larger cells at birth by arresting HeLa-hgem cells using Roscovitine, an inhibitor of major interphase cyclin dependent kinases, like Cdk2[48]. After a 48 h block with Roscovitine, the drug was rinsed, and cells were injected in the volume measurement chamber (Fig. 6a, Supplementary Movie 4). Cells which had been treated with Roscovitine were on average 1.7-fold larger than the controls (Fig. 6b, top histogram). Analysis of steadiness and homeostatic behavior in S-G2 is shown in Supplementary Fig. 6a-d as we focus here on control in G1. As expected, large Roscovitine-treated cells displayed a shorter G1 duration (Fig. 6b, right histogram) and were on average closer to a minimal G1 duration (about 4 h), independently of their volume at birth (Fig. 6b). Surprisingly, the large Roscovitine treated cells which had lost G1 modulation grew, during G1, by a constant amount of volume which was independent of their volume at birth and on average similar to that of the control condition (Welch $t$ test comparing the means, $p = 0.2423$) (Fig. 6c).

**Growth-rate modulations contribute to size correction**. If G1 duration is not modulated, an alternative mechanism to control size could be a modulation of growth rate. To assess the growth mode of cells in this experiment, we analyzed single cells growth curves in G1 and looked at how the instantaneous growth speed (i.e., the growth speed measured over short periods of time, $dt = $ 90 min) correlated with volume during this period of time (see Method and Supplementary Fig. 7a-c). This showed that, for both control and Roscovitine-treated cells, and for all the range of volumes, growth speed in G1 increased linearly with volume, compatible with an exponential growth mode even for the largest

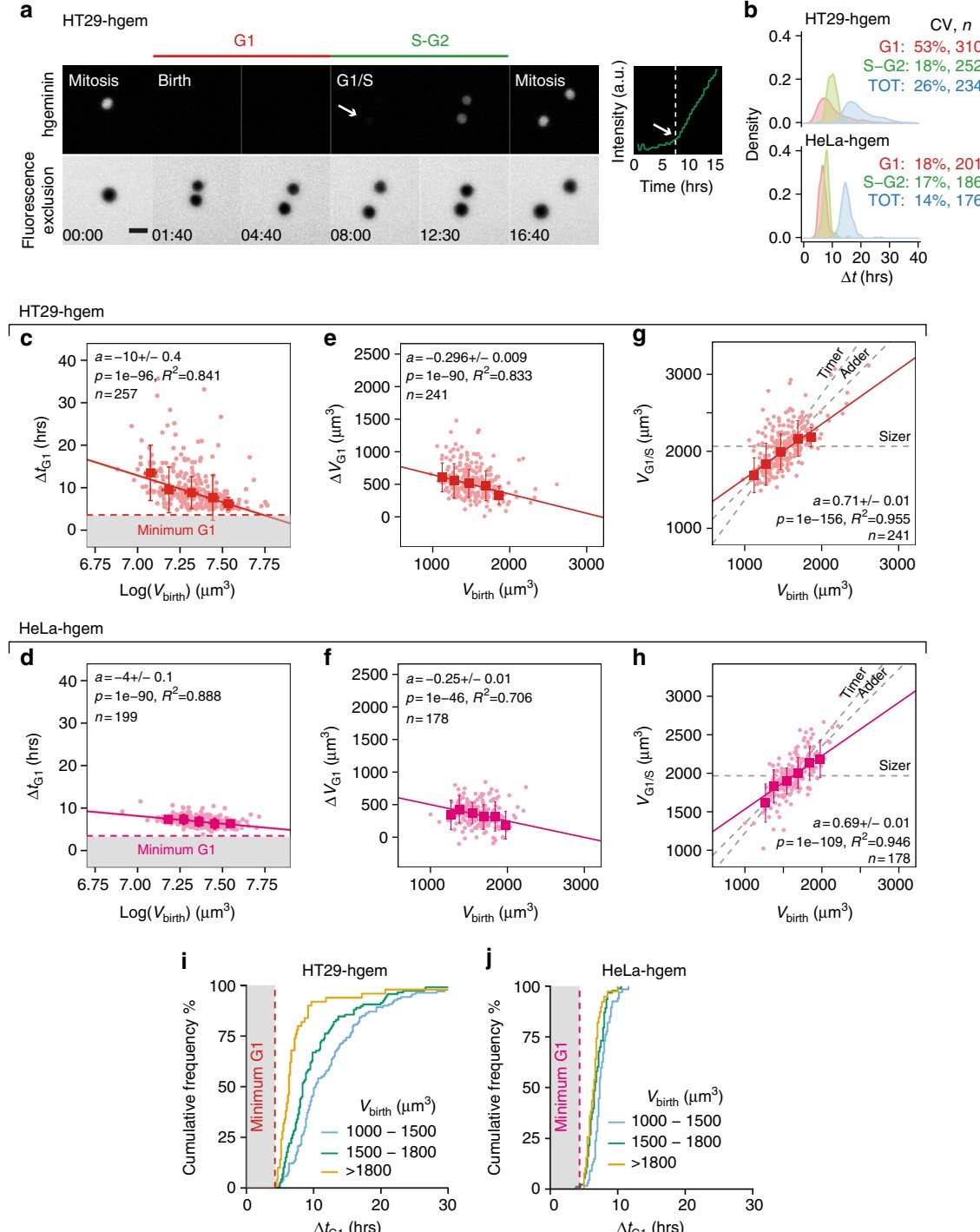

**Fig. 4** Modulation of G1 duration as a function of volume at birth. **a** Sequential images of HT29 cells expressing hgeminin-mcherry (top row) in an FXm chamber (bottom row). Right graph shows the quantification of hgeminin-mcherry in the cell as a function of time. Time zero corresponds to mitosis. The vertical white dashed line and arrows indicate the time at which hgeminin-mcherry becomes detectable. G1 phase (red line) spans from birth to appearance of hgeminin (G1/S transition) and S-G2 phases (green line) from G1/S to next entry in mitosis. Scale bar is 20 μm. Time is in hours:minutes. **b** Kernel density estimates of the duration $\Delta t$ of G1 phase (red), S-G2 phase (green) and total cell cycle (blue) for both HT29-hgem and HeLa-hgem. CV is the coefficient of variation (in %). **c**, **d** Duration of G1 phase, $\Delta t_{G1}$ as a function of the logarithm of volume at birth ($V_{birth}$) for HT29-hgem ($N = 4$) (**c**) and HeLa-hgem ($N = 2$) (**d**). Red dashed line and gray area are a visual guide for minimum G1 duration around 4 h. **e**, **f** Total added volume in G1 $\Delta V_{G1}$ as a function of volume at birth ($V_{birth}$) for HT29-hgem ($N = 4$) (**e**) and HeLa-hgem ($N = 2$) (**f**). **g**, **h** Volume at G1/S ($V_{G1/S}$) vs. volume at birth ($V_{birth}$) for HT29-hgem ($N = 4$) (**g**) and HeLa-hgem ($N = 2$) (**h**). The dashed gray lines indicate the expected trend in the case of a timer (slope $= \langle V_{G1/S}/V_{birth}\rangle$, intercept $= 0$), an adder (slope $= 1$, intercept $= \langle V_{G1/S}\rangle$) and a sizer (slope $= 0$, intercept $= \langle V_{G1/S}\rangle$). **i**, **j** Cumulative frequency graph of G1 duration binned for three ranges of volumes at birth $V_{birth}$ for HT29-hgem (**i**) ($N = 4$) and HeLa-hgem (**j**) ($N = 2$). Dashed line and gray area are a visual guide for minimum G1 duration around 4 h. For the plots in **c–h**, individual cell measures (dots) and median bins (squares) ± s.d. (bars) are shown. Solid lines are linear regressions on the median bins weighted by the number of observations in each bin. See also Supplementary Figure 4 and Supplementary Movie 3

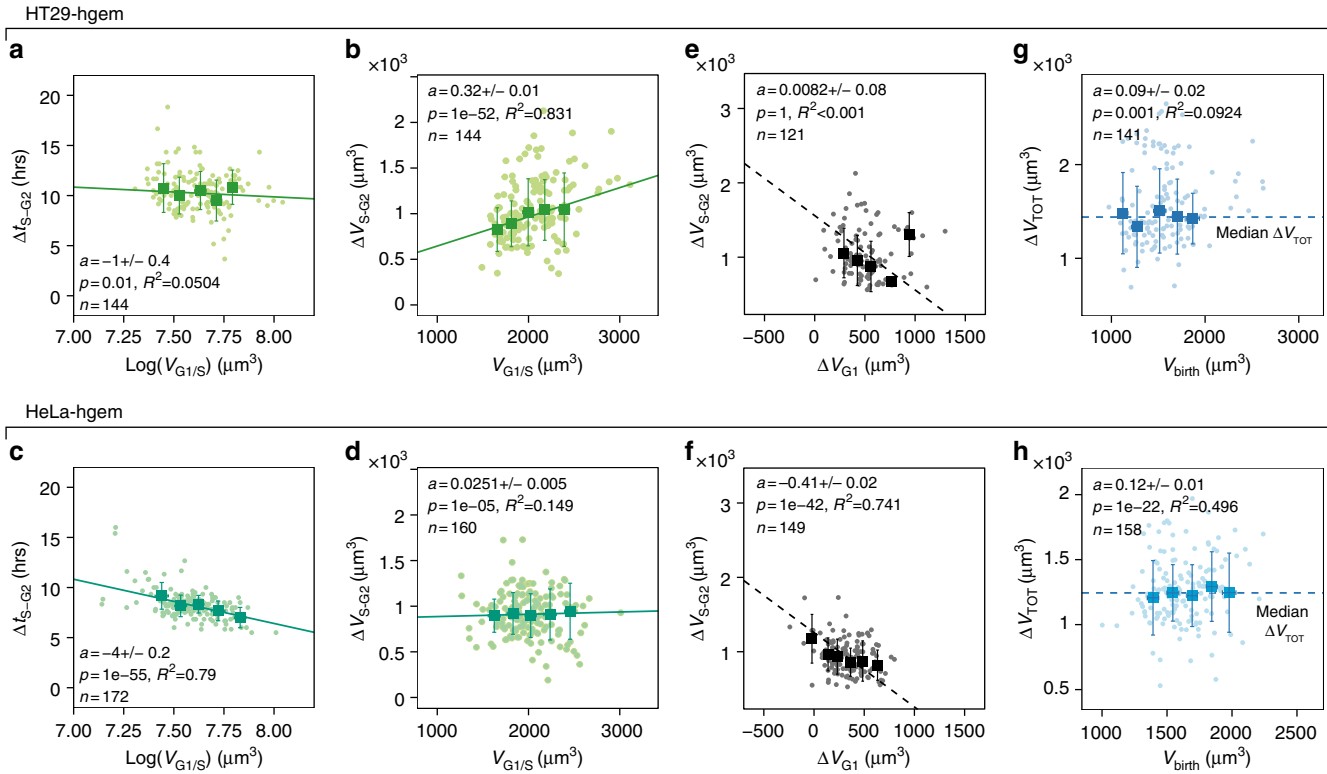

**Fig. 5** S-G2 duration is negatively correlated with volume at G1/S in HeLa but not HT29 cell. **a** Duration of S-G2 phase, $\Delta t_{S-G2}$ vs. the logarithm of volume at G1/S transition ($V_{G1/S}$) for HT29-hgem ($N = 4$). **b** Added volume in S-G2 phase, $\Delta V_{S-G2}$ vs. volume at G1/S transition ($V_{G1/S}$) for HT29-hgem ($N = 4$). **c** Duration of S-G2 phase, $\Delta t_{S-G2}$ vs. the logarithm of volume at G1/S transition ($V_{G-S}$) for HeLa-hgem ($N = 2$). **d** Added volume in S-G2 phase, $\Delta V_{S-G2}$ vs. volume at G1/S transition ($V_{G1/S}$) for HeLa-hgem ($N = 2$). **e, f** Added volume in S-G2, $\Delta V_{S-G2}$ vs. added volume in G1 ($V_{G1}$) for HT29-hgem ($N = 4$) (**e**) and HeLa-hgem ($N = 2$) (**f**). Dashed black line represents the slope expected in the case of a mechanistic adder where: $\Delta V_{S-G2} = \langle \Delta V_{TOT} \rangle - \Delta V_{G1}$ (slope of $-1$). **g, h** Added volume in the whole cell cycle $\Delta V_{TOT}$ vs. volume at birth ($V_{birth}$) for HT29-hgem ($N = 4$) (**g**) and HeLa-hgem ($N = 2$) (**f**). For all the plots in this figure, individual cell measures (dots) and median bins (squares) ± s.d. (bars) are shown. Solid line is a linear regression on the median bins weighted by the number of observations in each bin. See also Supplementary Figure 5

cells (Fig. 6d for G1, Supplementary Fig. 7d, e for S-G2 and complete cell cycle, and Supplementary Fig. 7f relative to G1/S transition). Thus, the growth modulation that leads to size control in large cells has to be more complex than a simple switch to a linear mode of growth.

To better characterize a potential growth rate modulation, we grouped Roscovitine and control cells and repeated the plot of instantaneous growth speed as a function of volume as in Fig. 6d but defined three sub-groups of cells containing: (i) the 20% smallest cells at birth, (ii) the intermediate-sized cells and (iii) the 80% largest cells at birth (Fig. 6e). We recall here that, by definition, the slope of such plot indicates the growth rate of cells. This analysis showed that although for all ranges of size at birth growth was compatible with exponential, the slope of growth speed vs. volume decreased for larger sizes at birth, suggesting a lower growth rate for cells born larger (Fig. 6e). This conclusion holds true even without the Roscovitine condition since the first two groups of cells (the 20% smallest and intermediate sized cells) contained a majority of cells from the control condition.

In conclusion, large Roscovitine-treated HeLa cells bring further evidence of a minimum G1 duration (Fig. 6b) already suggested by the results in control HeLa (Fig. 4d–j) and HT29 cells (Fig. 4c–i). Moreover, this experiment provides a direct example of cells for which modulation of the growth rate in G1 as a function of volume at birth can contribute to size control.

**Mathematical framework comparing size control across organisms.** Our results show evidence of time modulation in G1,

in agreement with recent findings[32,36] and directly support the hypothesis that modulations of the growth rate might also contribute to size homeostasis[24,25]. To understand the respective contribution of growth and time modulation to the effective homeostatic process, we built a general mathematical framework that allowed us to perform a comparative analysis of size homeostasis mechanisms in mammalian cells and unicellular organisms. Our model (described in details in Supplementary Note 1) assumes that cells grow exponentially, which corresponds to the average behavior we observed in our dataset, and adopt a rate chosen stochastically from a probability distribution. This rate may depend on volume at birth (and hence contribute to size correction). Similarly, cell cycle duration may be chosen based on volume at birth and has a stochastic component. Correlations between growth rate, cell cycle duration and size at birth are accounted to linear order, motivated by the fact that such linear correlations are able to explain most patterns in existing data (at least for bacteria[49]). The resulting model is able to characterize the joint correction of size by timing and growth rate modulation, with a small number of parameters.

A first parameter, $\lambda$, describes how the total relative growth ($\log(V_{mitosis}/V_{birth})$) depends on volume at birth. If $\lambda = 1$, the system behaves like a sizer, if it is 0.5, it is an adder and if it is 0, there is no size control at all (on average, cells divide when they doubled their initial volume). This parameter can be described, for each dataset, by performing a linear regression on the plot of $\log(V_{mitosis}/V_{birth})$ vs. the $\log(V_{birth})$ (Fig. 7a and Equation 5 in Supplementary Note 1). The second parameter, $\theta$, describes how

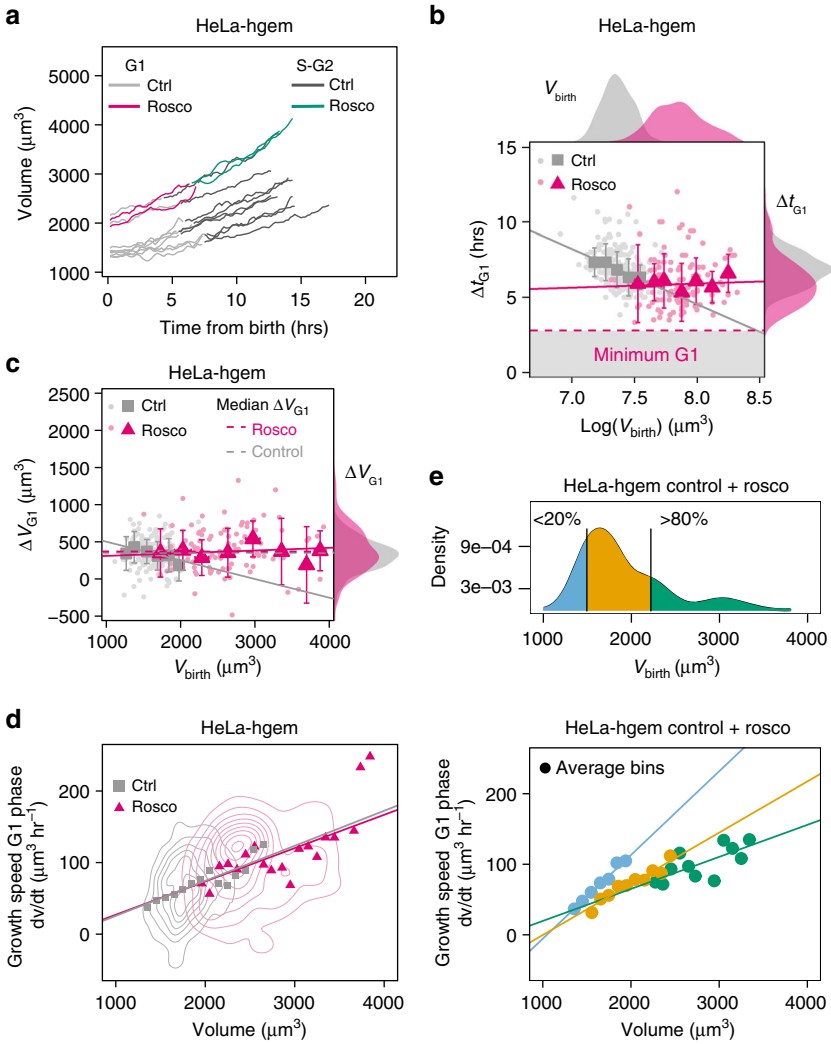

**Fig. 6** Size correction by growth-rate modulation in control and abnormal large Hela cells. **a** Examples of single-cell growth trajectories for HeLa-hgem cells, either control ('ctrl'), or after washout from Roscovitine treatment ('rosco') as a function of time from birth. **b** Duration of G1, $\Delta t_{G1}$ as a function of the logarithm of volume at birth ($V_{birth}$) for HeLa-hgem cells. Results from the linear fit: control: $a = -4 \pm 0.1$, $p = 1*10^{-90}$, $R^2 = 0.888$, $n = 199$, $N = 2$; Roscovitine: $a = 0 \pm 0.2$, $R^2 = 0.019$, $p = 1$, $n = 120$, $N = 3$. Red dashed line and gray area are a visual guide for minimum G1 duration. Top: kernel estimates of volume at birth; control: $\langle \log V_{birth} \rangle = 7.37$, $n = 231$, $N = 2$; Roscovitine: $\langle \log V_{birth} \rangle = 7.86$, $n = 136$; Welch $t$ test comparing the means: $p = 2.2 \times 10^{-16}$. Right: kernel estimates of $\Delta t_{G1}$; control: $\langle \Delta t_{G1} \rangle = 7.0$ h., $n = 201$, $N = 2$; Roscovitine: $\langle \Delta t_{G1} \rangle = 6.1$ h, $n = 124$, $N = 3$; Welch $t$ test comparing the means: $p = 6.5 \times 10^{-7}$. **c** Added volume in G1 ($\Delta V_{G1}$) vs. volume at birth for HeLa-hgem cells. Results from the linear fit: control: $a = -0.25 \pm 0.01$, $p = 1 \times 10^{-46}$, $R^2 = 0.706$, $n = 178$, $N = 2$; Roscovitine (red line): $a = 0.1 \pm 0.02$, $p = 0.1$, $R^2 = 0.046$, $n = 108$, $N = 3$. Dashed lines represent the median added volume in G1 for the control ($\langle \Delta V_{G1} \rangle = 350 \, \mu m^3$, $n = 178$) and the Roscovitine ($\langle \Delta V_{G1} \rangle = 390 \, \mu m^3$, $n = 108$) condition. Right: kernel estimates of $\Delta V_{G1}$. Welch's $t$ test comparing the mean added volume: $p = 0.2423$. **d** Instantaneous growth speed $dv/dt$ in G1 as a function of volume, with bivariate kernel densities (concentric circles) and average bins for control ($n = 119$, $N = 1$) and Roscovitine ($n = 49$, $N = 2$) conditions. Results from the linear fits, control: $a = 0.0489 \pm 0.0005$, $p \approx 0$, $R^2 = 0.78$; Roscovitine: $a = 0.047 \pm 0.002$, $p = 1 \times 10^{-137}$, $R^2 = 0.49$. **e** Top: kernel density of volume at birth for control and Roscovitine treated HeLa-hgem cells grouped together. Bars represent the 20 and 80% percentiles and define three groups: cells within the 0–20% percentile (blue), 20–80% percentile (orange) and 80–100% percentile (green). Bottom: Same data as **d** but for the three groups analyzed separately. Results from the linear fits (lines) on the average bins (dots) for each group with nc (number of control cells) and nr (number of Rocovitine-treated cells): 0–20%: $a = 0.119 \pm 0.008$, $p = 4.1 \times 10^{-5}$, $R^2 = 0.98$, nc = 24, nr = 0; 20–80%: $a = 0.072 \pm 0.009$, $p = 4.88 \times 10^{-5}$, $R^2 = 0.90$, nc = 60, nr = 15; 80–100%: $a = 0.05 \pm 0.01$, $p = 0.00192$, $R^2 = 0.43$, nc = 3, nr = 24. For **b**–**d**, control condition ('ctrl') is in gray and Roscovitine-treated condition ('rosco') is in red. Individual cell measures (dots) as well as median (**c**, **d**) or average (**d**) bins (ctrl: squares, rosco: triangles) and s.d. (bars) are shown. Solid lines shows linear regression on the bins weighted by the number of event in each bin. a is alsways given as slope ± standard error. See also Supplementary Figures 6 and 7, Supplementary Movie 4

cell cycle duration depends on volume at birth. This parameter can be described, for each dataset, by performing a linear regression on the plot of cell cycle duration ($\tau = \Delta T$) vs. log ($V_{birth}$) (Fig. 7b and Equation 6 in Supplementary Note 1). If this correlation is negative (which, by choice, corresponds to a positive value of the parameter meant to describe the strength of the correction), it means that larger cells will tend to divide in

shorter times, hence that modulation of timing contributes to size correction. Finally, the third parameter, $\gamma$, describes the link between initial size and a variation in growth rate with respect to its mean value. Similarly, if $\gamma$ is positive, modulations of growth rate positively contribute to size control (Fig. 7c, Equation 4 in Supplementary Note 1). $\gamma$ can be obtained by a linear regression when the corresponding measurements are available (in datasets

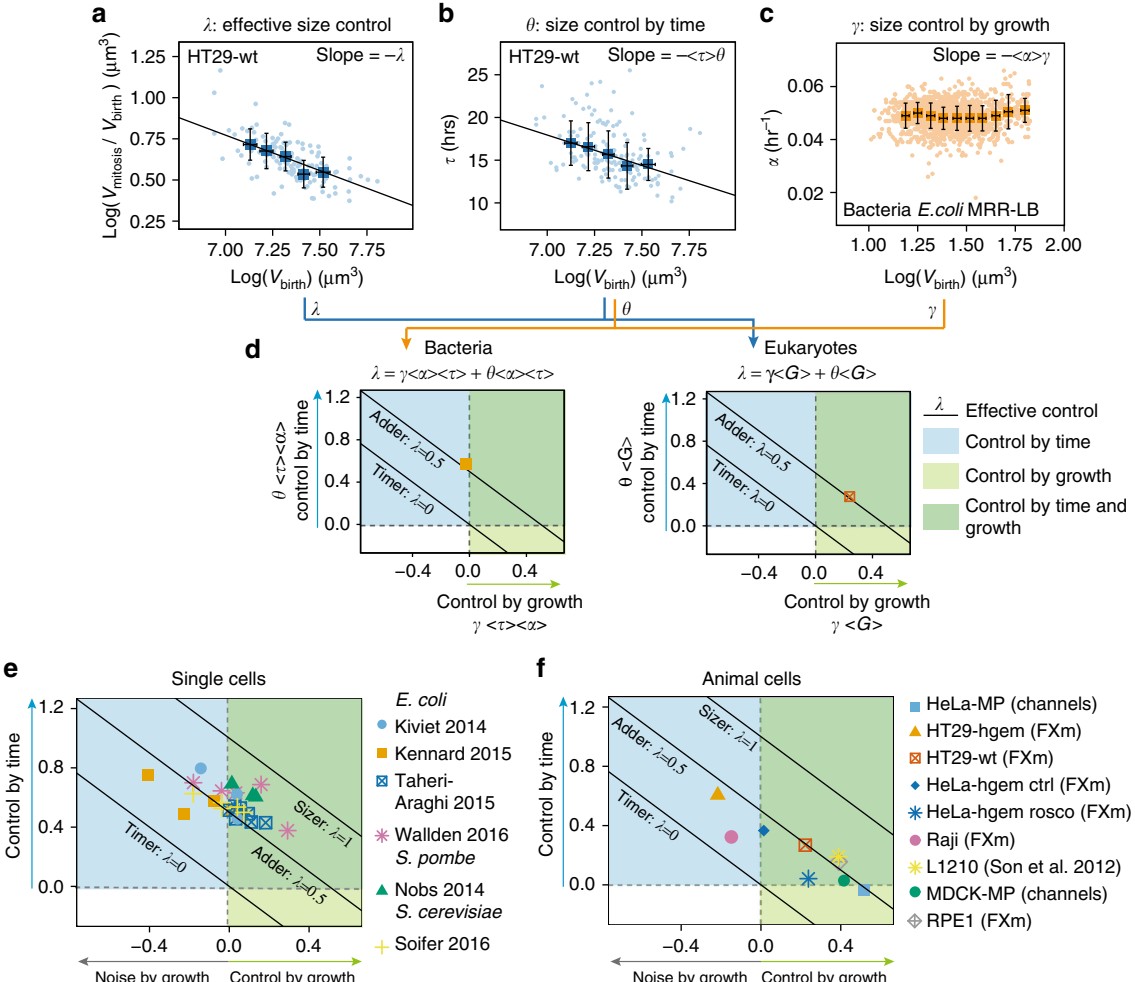

**Fig. 7** Contribution of growth and time modulation in overall size control. **a** Replicative growth, $\log(V_{mitosis}/V_{birth})$ vs. logarithm of volume at birth $\log(V_{birth})$ for HT29-wt cells. The slope coefficient of the linear regression gives $-\lambda$ and indicates the strength of the effective size control ($-\lambda = -0.5 \pm 0.002$, $R^2 = 0.85$, $n = 132$, $N = 3$). **b** Cell cycle duration $\tau$ vs. initial volume $\log(V_{birth})$ for HT29-wt cells. The slope coefficient of the linear regression gives $-\langle\tau\rangle\langle\theta\rangle$, with $\langle\tau\rangle$ the average cell cycle duration and $\theta$ the strength of control by time modulation. A positive value of $\theta$ corresponds to a positive effect on size control ($\langle-\tau\rangle\theta = -7 \pm 0.2$, $R^2 = 0.88$, $n = 163$, $N = 3$). **c** Growth rate $\alpha$ vs. volume at birth $\log(V_{birth})$, for a dataset on bacteria from ref.[51]. The slope coefficient of the linear regression gives $-\langle\alpha\rangle\langle\gamma\rangle$, with $\langle\alpha\rangle$ the average growth rate and $\gamma$ the control due to growth rate modulations. A positive value of $\gamma$ corresponds to a positive effect on size control ($-\langle\alpha\rangle\gamma = -0.0005 \pm 0.0002$, $R^2 = 0.06$, $n = 2107$). **d** Left: plot of $\theta\langle\tau\rangle\langle\alpha\rangle$, vs. $\gamma\langle\tau\rangle\langle\alpha\rangle$ for the bacteria dataset shown in Fig. 7c. Positive values along both y and x axes correspond to a positive effect on size control via time or growth modulation respectively. Right: plot of $\theta$ multiplied by $\langle G \rangle$, the average replicative growth $\langle G\rangle = \langle\log(V_{mitosis})/\log(V_{birth})\rangle$, vs. $\gamma$ multiplied by $\langle G\rangle$ for HT29-wt cells shown in **a** and **b**. **e** Comparison of datasets for bacteria (data from refs.[8,50–52]) and yeasts (data from refs.[11,16]), plotted as in **d**. Each point corresponds to a different growth condition (see Supplementary Fig. 8d). **f** Comparison of datasets for animal cells (our results and data from ref.[30].), plotted as in Fig. 8d. **a**, **b**, **c** Dots are single-cell measurements, squares with error bars are median bins with s.d., and black lines show the linear regression performed on the median bins weighted by the number of observations in each bin. **d**–**f** The dashed lines indicate the threshold above which time modulation (horizontal line) and growth modulation (vertical line) have a positive effect on size control. Values are given as slope ± standard error. See also Supplementary Figure 8

from bacteria[8,50–52]), or estimated from the values of $\lambda$ and $\theta$ for yeast and animal cells datasets where single cell growth rate was not available. The validity of this estimation was verified on the bacteria datasets (Supplementary Fig. 8a-b and Supplementary Note 1).

These three parameters are linked by a balance relation, which describes the fact that the overall size correction results from the combination of timing and growth rate corrections (see also Supplementary Note 1).

$$\lambda = \theta\langle\alpha\rangle\langle\tau\rangle + \gamma\langle\alpha\rangle\langle\tau\rangle \qquad (1)$$

Each cell line and condition can be characterized by one value for each parameter and thus one point on the graph which shows

$\gamma$ vs. $\theta$ (Fig. 7d). Additional (less relevant here) parameters concern the intrinsic stochasticity of cell cycle duration, growth rates and net growth (see Supplementary Information). For eukaryotes where the growth rate $\langle\alpha\rangle$ is not easily accessible, the product $\langle\alpha\rangle\langle\tau\rangle$ was approximated by:

$$\langle\alpha\rangle\langle\tau\rangle \approx \langle G\rangle = \langle\log(V_{mitosis})/\log(V_{birth})\rangle \qquad (2)$$

(Fig. 7d, right). The validity of this normalization was tested with bacteria (Supplementary Fig. 8c and Supplementary Note 1).

Using these dimensionless parameters, it was then possible to compare datasets obtained from different cell types in different conditions and estimate whether they displayed volume home-ostasis ($\lambda > 0$) with an adder behavior ($\lambda = 0.5$) or better ($\lambda = 0$).

It was also possible to know if homeostasis relied more on time modulation ($\theta > 0$) or growth rate modulation ($\gamma > 0$).

**Various couplings of growth and time modulations generate an adder**. With this framework, all the datasets for both bacteria[8,50–52] and yeasts[11,15] mostly fell around the line of $\lambda = 0.5$, indicative of a near-adder behavior (Fig. 7e and Supplementary Fig. 8d). Most mammalian cells also displayed volume homeostasis close to an adder behavior (all points except the Raji cells fell clustered around the line representing $\lambda = 0.5$, Fig. 7f), consistent with the plot shown in Fig. 2b. For both mammalian cells and bacteria, no dataset showed a negative time modulation, meaning that time modulation, when it is observed, always contribute to homeostasis. With comparison to yeast and bacteria, positive contribution of growth rate modulations to size control was stronger ($\gamma > 0$) and observed more often in mammalian cells. Negative growth rate modulation (larger cells with a faster exponential growth rate than smaller cells at birth), which was observed for some yeasts and bacteria, was also observed in two cases in mammalian cells (for Raji cells and HT29-hgem, Fig. 7f). Our analysis method, by providing a summarized overview of a large dataset comprising various cell types and culture conditions, demonstrated the generality of the phenomenological adder (or near-adder) behavior, and also revealed the diversity of the underlying homeostatic mechanisms with different coupling of growth rate and timing modulation. Such diversity was even observed in experiments on the same cell line depending on the growth conditions (datasets from bacteria) or initial size (results from Roscovitine-induced large HeLa cells).

## Discussion

The current understanding of size homeostasis in mammalian cells derives in large part from indirect evidence, due to experimental limitations. To tackle these limitations, we have developed FXm[38,39], a method that tracks the volume of individual mammalian cells over long periods of time, allowing direct measurements of freely growing and dividing cells. We show that the near-adder behavior is commonly observed in a variety of cultured and primary mammalian cells, similarly to yeast and bacteria. We provide direct evidence for a contribution of both growth rate and time modulation in size control and quantify their relative contribution in a general mathematical framework. Future work deciphering the molecular mechanisms of these adaptive modulations is required.

Our results on HeLa and HT29 cells confirm previous findings[32] implicating modulation of G1 duration in size control for mammalian cells, with a constraint on a minimal G1 duration above which large cells cycle in a minimal time, independent of initial size (Figs. 4c, d, i, j and 6b). In order to identify the molecular players of G1 size-checkpoint in mammalian cells, methods such as the FXm that enable single live cell size tracking will be a powerful tool to combine with reporters of recently identified key regulators of the G1/S transition[36,53,54]. S-G2 modulation was also observed in HeLa cells but not HT29 (Fig. 5a–c). This could reflect the existence of an additional size-checkpoint (similar to the 'cryptic' size-checkpoints in yeast[55]) observed in some cell types and not others, or observed when cells cycle very fast (like HeLa). Alternatively, it could be the sign of an over-arching control on cell size (the mechanistic adder)[45], as suggested by the negative correlation between added volume in S-G2 and added volume in G1 for HeLa cells (Fig. 5f).

Our dataset also provides direct evidence in support of a role for growth rate modulations in size homeostasis[24,27]. In particular, experiments on Roscovitine-induced abnormally large HeLa cells showed that such cells grew on average exponentially

(Fig. 6d and Supplementary Fig. 7d,e), did not adapt G1 duration to initial size (Fig. 6b), and yet maintained a size homeostasis behavior (Fig. 6c). This might be achieved through an adaptation of the exponential growth rate to the volume at birth (Fig. 6e). When considering single-cell growth trajectories, we observed that individual cells could display complex growth behaviors, with alternating plateaus and growth phases not clearly correlated with cell cycle stage events (Supplementary Fig. 7b-c). Only modulations that are size-dependent can impact cell size control, while general modulations, such as phase-dependent modulation of growth[56] (Supplementary Fig. 7f), do not contribute to size homeostasis. The factors that could modulate growth rate at the single-cell level in a size-dependent manner are as yet unknown. They could involve, as recently hypothesized, limitations of protein synthesis rate in large cells[57], nonlinear metabolic scaling with cell size[58,59], physical constraints on volume growth via the addition of surface area[60], or dynamic changes in cell/substrate adhesion, cell spreading, and cortical tension.

Our unbiased mathematical framework quantifies, for all cell types and all growth conditions, the respective contributions of growth and time modulation to the effective size homeostasis behavior (Fig. 7e, f). This analysis allowed us to compare the size homeostasis behavior of widely differing cells, and revealed global similarities, but also striking differences between mammalian cells and unicellular organisms. The adder behavior has been observed in a variety of unicellular organisms, from bacteria[7,8,11] to budding yeast[11,45] and we showed that this behavior is also very common in cultured and primary cells (Figs. 2b and 3b). However, the apparent universality of the adder at the phenomenological level may mask a more complex picture, where several regulatory mechanisms acting in parallel or sequentially might be at play[42]. Our mathematical framework shows that in bacteria, yeasts and animal cells, a variety of coupling between growth rate modulation and cell cycle duration modulation can lead to the same effective size control behavior. Second, within the group of cells we studied, growth rate modulation played a major contribution to size homeostasis in animal cells but less in yeast and bacteria (Fig. 7e, f). Environmentally dictated changes in growth rate are widely regarded as a central parameter for cell size homeostasis in multicellular organisms[2,61,62]. Thus, we surmise that the flexibility in patterns of growth may have to do with the acquisition of controlled and coordinated growth in tissues, which requires cells to respond quickly and efficiently to numerous simultaneous environmental cues. Understanding the physical parameters that drive animal cell growth in cell culture or in more complex tissue-like environment and combining this with the well characterized growth pathways[63] is a challenging and promising research question.

## Methods

**Reagents**. All the cell culture media were supplemented with 10% FBS and 1% penicillin–streptomycin. Media (#31053044, #21041025, #11875093, #61965026), EDTA, trypsin, penicillin–streptomycin, Insulin-Transferrin-Selenium-Sodium Pyruvate (#51300044), and glutamax (#35050061) were purchased from Thermo-Fisher. Zeocin (#10072492) was purchased from Life Technologies and Puromycin (#BML-GR312-0050) from Enzo life sciences. HeLa, MDCK, HT29 were cultured in DMEM-Glutamax and imaged in a media of the same composition but without phenol red. RPE1 and primary NHDF cells were cultured and imaged using DMEM-F12. Raji cells were cultured and imaged in RPMI-1640 supplemented with Glutamax. NAF cells were cultured and imaged in DMEM, no-phenol red, supplemented with Glutamax and Insulin-Transferrin-Selenium-Sodium Pyruvate. Dextran (#D-22910, #D-22914, #FD10S) and Roscovitine (#R7772-1G) were purchased from Sigma Alrich. The stock solution of dextran was 10 mg mL$^{-1}$ in PBS, the stock solution of Roscovitine was 50 mM in DMSO.

**Cell lines and plasmids**. HeLa cells are human cancerous epithelial cells from adenocarcinoma. HeLa expressing hgeminin-GFP (HeLa-hgem) were a kind gift from Buzz Baum's lab (UCL, London, United Kingdom). HeLa Kyoto expressing MyrPalm-mEGFP-H2B-mRFP (HeLa-MP) are a kind gift from Daniel Gerlich's

lab (ETH, Zurich, Switzerland). HT29 cells are human cancerous cells coming from colorectal adenocarcinoma. HT29 wild type cells (HT29-wt) were HT29 HTB-38 bought from ATCC. A stable HT29 cell line expressing hgem-mCherry (HT29-hgem) was established using the lentiviral vector mCherry-hGeminin(1/60)/pCSII-EF:[64] electroporation was used to transfect the cells, the cells were then selected with zeomycin 200 µg mL$^{-1}$ and FACS-sorted for mCherry fluorescence. The resulting polyclonal population showed a good homogeneity in fluorescence intensity. MDCK cells are dog epithelial cells from an apparently normal kidney. They are however hyperdiploid with a modal chromosome number ranging from 77 to 80 or 87 to 90 (instead of 78 for this specie). MDCK cells were obtained from Buzz Baum lab (UCL, London, United Kingdom). Similarly to the protocol used for HT29 expressing hgem-mcherry, a stable MDCK cell line expressing MyrPalm-GFP (MDCK-MP) was established by electroporating cells with the plasmid pMyrPalm-mEGFP-IRES_puro2b offered by Daniel Gerlich's lab. Selection was made with Puromycin 2 µg mL$^{-1}$ prior to FACS sorting. For all the transfected cell lines, antibiotic were removed from the culture media after FACS sorting. Raji cells are human B lymphoblastoid cells coming from a lymphoma. Raji were obtained from Claire Hivroz's lab (Institut Curie, Paris, France). RPE1 cells are human retinal pigment epithelial cells and were a kind gift from Anne Paoletti's lab (Institut Curie, Paris, France). Cell lines were tested for Mycoplasma every 6 months approximately and the tests were always negative.

**Extraction and culture of primary cells**. NHDFs are primary cells extracted from human abdominal skin and were bought from Biopredic. NAFs were a kind gift from Danjiela Vignevic's lab, (Institut Curie, Paris, France). They are human primary fibroblasts isolated from fresh healthy intestinal tissue of patients with locally advanced rectal cancer. Sampling protocol was approved by the designed ethics committee (CPP, Comité de Protection des Patients) and all patients gave written informed consent. The three types of NAFs used in this paper come from two different patients (for one patient, two samples (NAF-A and NAF-C), extracted at two distinct locations were taken). The protocol for sample collection and preparation is described in refs. [65,66]. Briefly, samples were collected after surgical resection in DMEM medium supplemented with 1% Antibiotic-Antimycotic. Tissue was mechanically resected in 1 mm piece, plated on scratched 10 cm Petri dishes and cultured in DMEM supplemented with 10% FBS, 1% Insulin-Transferrin-Selenium (ITS) and 1% Antibiotic-Antimycotic at 37 °C. Medium was changed every 3 days until fibroblasts emerged from the tissue peace. At this time cells were trypsinized and cultured under normal conditions for up to 10 passages.

**Microchannel experiments**. Microchannels molds were made with classical lithography technics and then replicated in epoxy molds. Microchannels had a 104 µm$^2$ cross-section area (13 µm width by 8 µm height) (Supplementary Fig. 1a). They were crossed perpendicularly by two large distributing channels (5 mm width by 50 µm height). The microchannels chips were replicated in PDMS, plasma-treated, bound to glass-bottom fluorodishes, coated with fibronectin 50 µg mL$^{-1}$ and incubated over night with the culture media. The large distributing channels were used both to inject the cells and as reservoirs of media. Cells were injected at a concentration of 3.8×10$^6$ mL$^{-1}$, in the upper distributing channel; the dishes were then tilted with the distribution channel up to depose cells at the entry of the microchannels by gravity. The opposing distributing branch contained only media and thus diffused nutrients to the channels. This was indeed important to guarantee enough nutrient stock and good growth conditions throughout the 50 h of the acquisition in this confined design. Cells were then let to migrate in the microchannels over-night and experiments were started the morning after.

Upon mitotic entry, cells round-up and adopt a cylinder shape because of the confinement. The contours of the cells were visualized by imaging of the protein MyrPalm-GFP to label cell membrane. Volume was calculated by measuring the length ($\ell$) of the cell and multiplying it by the channel cross section area (CS): $V \approx \ell.CS$. (Supplementary Fig. 1a). For the analysis, mitosis was defined as the first time-point where the cell rounds-up and displays a cylinder shape and birth was the last time point after cytokinesis where the cell is still in the shape of a cylinder (Fig. 2c). In the channels, cells cycled slower (their average cell cycle duration was, for HeLa-MP: mean = 24.9 h and for MDCK-MP: mean = 19.5 h, by comparison, HeLa-hgem have an average cell cycle duration of 16.2 h in a culture dish (Supplementary Fig. 1b)). They also showed indirect evidence of linear growth (large and small cells added the same amount of volume in the same amount of time, (Supplementary Fig. 2a, c), a behavior reminiscent of what has been observed in another study using microchannels[32]. However, we checked that volume at birth and average growth speed were constant through time in the experiment (Supplementary Fig. 1g), thus meaning that these experiments were performed in stationary conditions and constitute a valid dataset for our study.

**Volume measurement with FXm**. The FXm method was initially described in ref.[39] and a detailed protocol is available in ref.[38]. In these two previous works, we provide a number of controls to show that this method enables accurate measurement of volume independently of cell shape (e.g., cells that were measured before and after detaching from the substrate had the same volume as they became round). Volume measurement can be affected if cells uptake the fluorescent probe

(thus leading to an underestimation of the volume). To check that this was not the case, we plotted volume at birth though time in the experiment for all the cell types studied (Supplementary Fig. 1g and Supplementary Fig. 6a). We could confirm that, for all cell types except HeLa cells, volume at birth was steady throughout the experiment. For HeLa cell, we could see some uptake of the fluorescent probe (see Supplementary Movie 1) but the decrease in volume from the beginning to the end of the experiment (after 40 h) is below 10% and thus does not impact the analysis we perform in our work.

Except for Raji experiments, the design of the volume measurement chamber (Fig. 1a) included two side reservoirs that diffused nutrients to cells in the middle of the chamber through microchannels. Side reservoirs were 400 µm high and diffusion to the observation part was achieved through a grid of channels ($w =$ 100 µm, $l = 300$ µm, and $h = 5$ µm). The height of the chambers was ranging from 20 to 24 µm (depending on the chambers) for HT29 and HeLa cells, 15.5 or 18.2 µm for the Raji cells and 18.4 µm for RPE1, NAFs and NHDF cells.

A detailed protocol for the FXm experiment is available in ref.[38]. Briefly, the day before the experiment, chambers were replicated in PDMS (crosslinker:PDMS, 1:10). To prevent dextran leakage outside the chambers, the height of the inlets was risen by sticking 3–4 mm high PDMS cubes on top of each inlet, then 2 mm diameter punches were made for every inlets. Chambers were then irreversibly bounded to 35 mm diameter glass-bottom fluorodishes by plasma treatment. Finally, they were coated with fibronectin 50 µg mL$^{-1}$ (all cell types except RPE1) or 10 µg mL$^{-1}$ (RPE1), rinsed and then incubated overnight with the appropriate phenol-red-free media. During the acquisition, the chambers were covered with media to prevent desiccation through the PDMS and subsequent changes of the osmolarity of the media in the chamber. To prevent potential sources of variability in the growth speed or doubling rate caused by different proliferative states in the population, cells were cultured in controlled conditions prior to experiments and then seeded at constant concentration two days before starting the experiment (1×10$^5$ cm$^{-2}$ for HT29 cells and 1.9×10$^4$ cm$^{-2}$ for HeLa). Cells were detached using trypLE (Thermofisher #12605036) (all cell types except HeLa) for 5 min or less or EDTA (Life Technologies #15040–033) (HeLa) for 15–20 min to avoid cell aggregates and optimize adhesion time to the glass-bottom, fibronectin-coated chamber. Cells were injected in the central part of the chamber (Fig. 1a) at a concentration ranging from 1.5 to 2×10$^5$ cells per mL in order to obtain the appropriate density in the chambers using a narrow 10 µL pipet tip (HT29, HeLa, RPE1) or a 2 mL syringe (NAFs, NHDF). For adherent cells (all cell types except Raji cells), 4 h after seeding, media was changed with equilibrated media containing 1 mg mL$^{-1}$ of 10 kDa Dextran. Raji cells were injected together with the Dextran. The dextran used was 10 kDa Dextran coupled either to Alexa-645 (HeLa-hgem experiments), Alexa-488 (HT29-wt, HT29-hgem, Raji experiments) or FITC (RPE1, NHDF, NAFs experiments). Imaging started 2–4 h after changing the media to give time for media to equilibrate in the chamber and avoid possible inhomogeneity of dextran just after injection.

**Controls for FXm experiments on cultured cell lines**. We checked that cell cycle time was similar inside and outside the measurement device (Supplementary Fig. 1b). All the cancerous adherent cell types (HeLa-hgem, HT29-wt, and HT29-hgem) showed a slightly higher duration of cell cycle outside the device compared to inside the device. For RPE1 cells, the difference in cell cycle duration was not statistically significant although they seem to be cycling slightly faster outside than inside the FXm device. Suspended Raji cells on the contrary showed a slightly higher cell cycle duration inside the FXm device. There can be multiple reasons for this (i.e., higher concentration of proliferative signals secreted by the cells in the FXm chamber than in the large volume of the culture dish; larger access to oxygen in microfluidic devices, made of a gas permeable elastomer, PDMS, than at the bottom of a Petri dish, where cells can easily find themselves in hypoxic conditions). Overall, the difference in average cell cycle duration outside or inside the device was significant but small and this control shows that cells cycle on normal times in the FXm device.

**Controls for FXm experiments on primary cells**. Primary cells, which are samples directly coming from human patients and are known to progressively stop dividing in culture are overall more heterogeneous in culture than immortal cell lines. The comparison of the coefficient of variation of cell cycle duration or volume at birth of our four datasets on primary cells with that of four immortal cell lines (RPE1, HT29-wt, Raji, and L1210 from ref.[30].) illustrates the higher variability observed in primary cells populations (Supplementary Fig. 3a-b). Moreover, the change of culture environment, from the cell culture dish to the FXm chamber caused a change in the way cells grew, with a low overall replicative growth: the ratio $\langle V_{\mathrm{mitosis}}/V_{\mathrm{birth}} \rangle$ was about 1.5–1.6 (Fig. 3a). This is lower than the values we report for immortal cell lines (Fig. 3a, gray area, Supplementary Fig. 2d).

To check that this decrease in volume was not due to an uptake of the fluorescence probe we use for the FXm method, we compared the images and results with that of HeLa-hgem cells. For HeLa-hgem cells, the uptake of dextran was visible by eye, producing clear dots in the cells (Supplementary Fig. 3d) and led to a decrease in volume throughout the experiment that was below 10% (Supplementary Fig. 1g) and a ratio $\langle V_{\mathrm{mitosis}}/V_{\mathrm{birth}} \rangle$ equal to 1.8 (Supplementary Fig. 2d). In primary cells, we could not identify by eye any uptake of the fluorescent probe (Supplementary Fig. 3e), yet the ratio $\langle V_{\mathrm{mitosis}}/V_{\mathrm{birth}} \rangle$ was lower than that of

HeLa (from 1.5 to 1.6 depending on the experiments). We also checked that the average growth speed was constant throughout the experiments (Supplementary Fig. 3f), indicating that the decrease in size was not caused by a progressive decrease of average growth speed during the course of the experiment (induced for example by a repetitive illumination or a progressive depletion of nutrients or any other time-dependent parameter of the set-up). Unfortunately, it is not possible with our device to perform longer experiments in order to check whether primary cells reached a new steady state of size after a few generations.

Altogether, these controls allow us to eliminate a number of potential experimental bias that could have explained the decrease in size in these primary cells. Because of these differences, we did not include primary cells to our final mathematical framework in Fig. 7. However, the separate analysis of their homeostatic behavior reveals that they display an adder (NAFs) or near-adder (NHDF) (Fig. 3b) that involves very little modulation of cell cycle timing (Supplementary Fig. 3g-j).

**Choice of key time points for the FXm analysis**. For the analysis of the relationships between volume at birth, volume at mitosis and volume at G1/S and the duration or volume gained between two of these time points, cells were manually tracked. During mitosis, an abrupt and reversible increase of volume has been described previously by us and others[39,67] (Supplementary Fig. 1d). To make sure that we measured volume at mitosis and volume at birth outside from this mitotic volume overshoot, we defined mitosis as the point occurring 60 min prior to cytokinesis and birth as the point occurring 40 min after cytokinesis (Fig. 1b and Supplementary Fig. 1d). To check that volume at mitosis was measured before the mitotic volume overshoot, we compared the volume measured 100 min and 60 min before cytokinesis and verified that they were not significantly different (pairwise $t$ test comparing the means: $p = 0.800$) (Supplementary Fig. 1e). On average, the volume of the mother cell 60 min before mitosis is slightly higher than the sum of the volume of the two daughter cells at birth (mean: $m = 3200\ \mu m^3$ and $m = 2900\ \mu m^3$, respectively). The potential overestimation of volume at mitosis however remains below 10% of the average volume at mitosis and thus will not have an effect on the correlations studied. For the measurement of volume at birth 40 min after cytokinesis, we checked that segmentation of the daughter cells close to each other did not introduce mistakes in volume measurement. To do so, we compared the sum of the volumes of the two daughter cells measured separately with the value obtained when measuring the two cells at once (Supplementary Fig. 1e). These two measurements were not significantly different (pairwise t test comparing the means $p = 0.826$).

G1/S was identified as the first time-point where hgeminin-GFP (HeLa) or hgeminin-mcherry (HT29) was observed (Fig. 4a). This point was visually assessed by looking at the movies and we checked that this method was correct compared with an assessment from the fluorescence expression profile (Supplementary Fig. 4a). We compared 10 curves and show here the most imprecise evaluation (Supplementary Fig. 4a left), the average type of error observed (Supplementary Fig. 4a middle) and the best evaluation (Supplementary Fig. 4a right). This empirical check shows that on average the error was very small.

**Roscovitine experiment**. For the Roscovitine experiments, HeLa cells were seeded in six-well plates at $1.9\times10^4$ (control) and $8.3\times10^4$ (treated) cells per $cm^2$ 52 h before the experiment. Four hours later (48 h before the experiment), when cells were spread, media was changed to 2 mL ± 20 μM Roscovitine. Roscovitine stock solution was 50 mM in DMSO.

**Live-cell imaging**. Phenol red-free media was used for FXm experiments. Acquisitions were performed on a Ti inverted (Nikon) or Axio Ob-server microscope (Carl Zeiss) or DMi8 inverted microscope (Leica) at 37 °C with 5% $CO_2$ atmosphere, a ×10 dry objective (NA 0.30 phase) for FXm experiments or a ×20 dry objective (NA 0.45 phase) for microchannels experiments. Images were acquired using MetaMorph (Molecular Devices) or Axio Vision (Carl Zeiss) software. The excitation source was systematically a LED for FXm experiments to obtain the best possible homogeneity of field illumination (Lumencor or Zeiss Colibri); or a mercury arc lamp for some of the microchannel experiments. Images were acquired with a CoolSnap HQ2 camera (Photometrics) or an ORCA-Flash4.0 camera (Hammamatsu).

For time-lapse experiment, images were acquired every 5 min (microchannel experiments), 10 min (FXm measurements: fluorescence-exclusion channel and phase channel for HeLa, HT29, and Raji cells), 15 min (FXm measurements: fluorescence-exclusion channel and phase channel for RPE1, NAFs, and NHDF cells) and 30 min (fluorescent geminin channel) for up to 50 h in order to obtain 1–2 full cell cycles per lineage. One of the crucial parameter to preserve a good cycling of the cells in the FXm chambers throughout the 50 h of experiment is to reduce the power of the fluorescence lamp to the maximum. A useful landmark to adapt the parameters on different microscopes, was to set the power of the lamp in order to obtain around $2^{13}$ gray levels with excitation times of about 300–400 ms.

**Image analysis**. For FXm experiments, image analysis was performed using a home-made Matlab program described in ref.[39]. The growth curves were analyzed with an updated version of this program written in collaboration with the company QuantaCell[38]. Briefly, fluorescent signal was calibrated for every time points using the fluorescence intensity of the pillars and around the cell of interest to obtain the linear relationship between height and fluorescence. After background cleaning, the fluorescence intensity was integrated for the whole cell and its surroundings to obtain the cell volume.

For the microchannels experiments, image analysis was performed on ImageJ.

**Data filtering and analysis**. For all the data on animal cells (ours and from ref.[30].), only clear outliers that were higher or lower than the mean ± 3×s.d. (standard deviation) were removed. This corresponded on average to 0 to 5 points maximum per dataset (each dataset being $n > 87$). These outliers were removed for visual purposes (scale of the plot adapted to the range of the data) and analytical robustness.

For the bacteria and yeast data obtained from previous studies[8,11,15,51,52], a filter based on the IQR (interquartile range) was performed: cells for which $\log(V_{birth})$ and $\log(V_{mitosis})$ were higher or lower than 1.5×IQR ± median of $\log(V_{birth})$ and $\log(V_{mitosis})$, respectively were removed.

The growth curves were obtained from automated tracking of the movies and analyzed as follows. First, all the tracks were visualized to identify the phases in the cell cycle (birth was automatically detected because the tracks split when the newborn cells separated, mitotic volume overshoot indicating the end of the cell cycle was visually assessed from the volume growth curve (see Fig. 1c) and G1/S transition was visually assessed as the transition point in the nuclear-hgeminin fluorescence expression curve as shown in Supplementary Fig. 4a). Both complete cell cycle trajectories and incomplete trajectories that were longer than 5 h and contained at least one identified cell cycle event (birth, G1/S or mitosis) were kept. Second, clear outliers caused by errors of segmentation were removed using a sliding filter that removed a point if it were too far from the median of the local distribution of measures (on a window of 11 frames). This filter was good enough to remove only clear outliers (Supplementary Fig. 7a, left). Third, for the instantaneous growth speed measurement, the volume curves were smoothed by performing sliding average on windows of 7 frames (70 min). Then the growth speed at each point was the slope of a robust linear fit performed on windows of 9 frames centered on this point (Supplementary Fig. 7a, middle).

**Statistical analysis**. All the figures and statistical analysis were performed in R. Packages used were: "robust", "robustbase","ggplot2","grid","gridExtra","xtable","stringr","RColorBrewer".

For the boxplots, the upper and lower hinges correspond to the first and third quartiles, the upper and lower whiskers extend from the hinge to the highest (lowest) value within 1.5*IQR (Inter Quantile Range) of the hinge. Data beyond the whiskers are shown as outliers.

For the plots where a linear relationship was tested, a linear fit on the median bins, weighted by the number of observed variables in each bins was performed. The results of this fit is always indicated with the slope coefficient ($a$) ± its standard error, the $p$-value of the slope coefficient ($p$) and the coefficient of determination ($R^2$). For all the plots except the ones analyzing growth speed as a function of time or size (Fig. 6d, e and Supplementary Fig. 7d-f), the bins are median bins along the $x$ axis of the plot, and the bars represent the standard deviation. Equally spaced bins were defined along the $x$ axis and bins that contained less than a minimum number of single-cell events were removed. The bin number ($binn$) and the minimum number of events per bin ($minn$) was adapted to the size of the datasets as follows. For animal cells, the size of the datasets ranged from 80 to 300 oservations, $6{\leq}binn{\leq}8$ and $2{\leq}binn{\leq}8$. For bacteria or yeast datasets, $binn$ and $minn$ depended on the number $n$ of observation in the dataset: $n{<}100$, $binn = 8$, $minn = 8$; $100{<}n{<}1000$, $binn = 10$, $minn = 15$; $1000{\leq}n{<}5000$, $binn = 13$, $minn = 60$; $n{>}5000$, $binn = 15$, $minn = 150$. For the plots testing the relationship between growth speed and volume or time (Fig. 6d, e and Supplementary Fig. 7d-f), the bins are average bins and bins that contained measurements on less than five different cells were removed to avoid low-sampling effects.

**Data availability**. The authors declare that all data supporting the findings of this study are available within the article and its supplementary information Files or from the corresponding author upon reasonable request. The Matlab home-made software developed for volume measurement is available upon request.

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

## Acknowledgements

We thank Jorge Barbazan for helping with the experiments on NAFs, the Gerlich lab for sharing the HeLa-MP cell line, Helen K. Matthews, Nunu Mchedlichvili, and Ewa Zlotek-Zlotkiewicz, members of the Piel lab and the Perez lab for scientific and technical advices, Camille Blakeley and Charlotte Pirot for preliminary works as undergrad students, Tom Wyatt, Youmna Attieh, and Giuliana Victoria for help on revising the manuscript, Isabel Brito for advices on the statistical analysis, Laurence Bataille for help with the Raji cells, Lucie Sengmanivong and the imaging platform from the Institut Curie PICT-IBiSA, the UMR 168 clean room facility and the IPGG platform. We also acknowledge Jan

Skotheim for critical reading of the manuscript, Sungmin Son, Scott Manalis, Ariel Amir and Sander Tans for sharing and discussing data on yeast and bacteria. CC acknowledges support from the Fondation pour la Recherche Médicale (FDT20160435078) and the Ligue Nationale contre le Cancer for funding. M.C.L. acknowledges support from the International Human Frontier Science Program Organization, grant RGY0070/2014. B.B. acknowledges Cancer Research UK program grant for support: C1529/A17343. This work was supported by a LABEX IPGG grant to R.A., by an ERC consolidator grant (311205 PROMICO) to M.P., by an ANR grant to M.P. (ANR-14-CE11-0009-03, Cell-Size), by the Institut Pierre Gilles de Gennes (Equipement d'Excellence, " Investissements d'Avenir", program ANR-10-EQPX-34).

## Author contributions

C.C. and S.M. conducted the experiments, S.M. optimized and designed the FXm chambers, N.S. and P.S. performed some of the data analysis, E.T. and R.A. designed, produced, and characterized the molds for the chambers, M.C.-L. and J.G. developed the theoretical framework, B.B. helped conceive of project, helped supervise early part, helped with text, C.C. conducted the analysis, M.C.L. helped with data analysis, C.C. and M.C.-L. helped with the manuscript preparation, C.C. and M.P. designed the experiment, M.P. wrote the paper and supervised the work.

## Additional information

**Competing interests:** The authors declare no competing interests.

