## [Peer Review File · Nature Communications]

Reviewer #1 (Remarks to the Author):

The authors have carried out extensive amounts of additional experiments which significantly strengthen their paper. Also, the modifications to the manuscript text has improved the clarity of the work significantly. Most of the concern I had about this paper have been addressed. The only unaddressed concern is the mechanism behind the cell type differences in size control. While answering this question will be critical for the field, it is clearly beyond the scope of this work, and should not be required for publication. I therefore support the publication of this paper in Nature Communications.

--

Reviewer #2 (Remarks to the Author):

I find the revised manuscript to be significantly improved, and have only minor comments for additional changes. While the study may not "solve" the size control question in mammalian cells, it provides a wealth of solid data and interesting discussion that will aid the field in its long-term goals.

Minor comments:

1. When discussing figures 4e-f on page 8, I would encourage the authors to avoid interpreting the "generally negative" slope. This trend is extremely weak, to the point of being largely inconclusive, at least for me. By interpreting these weak data, the authors lose ground on their interesting interpretation of stronger data throughout the manuscript.

2. I have two suggested text changes for discussing the new data on primary cells on page 29 of the supplemental material. First, I would change "inhomogeneous" to "heterogeneous" on line 2. Second, the text on this page refers several times to "<Vbirth / Vmitosis>" when I believe they mean to indicate the reverse (i.e. <Vmitosis / Vbirth>).

--

Reviewer #4 (Remarks to the Author):

In this manuscript, Cadart et al. study the mechanisms of size control in several cell lines using FXm (fluorescence exclusion measurement). This method has the advantage of allowing the tracking of individual cells from birth to division in a controlled environment. Using this methodology, the authors describe how different cell types (and even subgroups of cells of the same cell type) can differently adapt the duration of G1 and S-G2 phase and their growth rate in order to achieve an effective near-adder behavior.

They conclude that while a phenomenological adder is commonly found in various cell types, there is not a universal underlying mechanism. These findings are integrated in a mathematical model that allows the comparison of size homeostasis behavior in different datasets.

The authors had submitted a previous version of the manuscript for publication in Nature Cell Biology, and several concerns were raised. Since this first submission the authors have added new experimental data and have addressed the main criticisms posed by the reviewers.

In general terms, I regard the quality of the experiments as high and I think that the data generated will be valuable to the field of size control. Hence, I recommend its publication in Nature Communications.

There are just a few points that I would like the authors to clarify/comment on:

1.- In the FXm experiments in supplementary figure 2, HT29-hgem cells contain a small sub-population with a longer G1 phase that add a larger volume over the cell cycle. In the figure the authors show the statistical analysis of the data once these outliers have been removed. However, it is unclear whether the same sub-population is also observed in subsequent experiments where this cell line is used (figures 3 and 4) (I assume it is), and, if so, whether they have been considered or excluded in the analysis. Do these cells with an abnormally long G1 phase modulate the length of S-G2?

2.- One of the points that the authors touch upon is the question of whether cell growth is linear or exponential. To address this matter, they analyze growth rate of HeLa cells (the slope of the plot of instantaneous growth speed as a function of volume). By grouping cells in three different categories according to their size they observe that, while in all cases there is a linear increase in growth speed as volume increases, the slope of the plot decreases for larger sizes at birth. Since the grouping of cells is arbitrary (20% smallest, 60% intermediate and 20% largest) I wonder whether a narrower grouping (in 5-10% steps) would show that the largest cells do not increase their growth speed as their volume increases, which would be indicative of linear growth after reaching a certain size.

3.- The advantage of the FXm method used throughout the manuscript is that cells grow without the space restriction of the microchannels. The authors argue that in the microchannels cells show indirect evidence of linear growth, whereas they observe exponential mode of growth with the FXm device. Nevertheless, in both cases size homeostasis seems to be achieved through an adder.

Although cells are undoubtedly happier in the FXm setup than in the microchannels, it is unclear whether it reflects the natural mode of growth of cells in a tissue (where growth might be confined by neighboring cells). Have the authors considered analyzing cell growth in confluent areas of the FXm device or after seeding a higher density of cells?

Response to referees

Reviewer #1 (Remarks to the Author):

The authors have carried out extensive amounts of additional experiments which significantly strengthen their paper. Also, the modifications to the manuscript text has improved the clarity of the work significantly. Most of the concern I had about this paper have been addressed. The only unaddressed concern is the mechanism behind the cell type differences in size control. While answering this question will be critical for the field, it is clearly beyond the scope of this work, and should not be required for publication. I therefore support the publication of this paper in Nature Communications.

Answer: We are grateful to the referee for his/her positive assessment of our manuscript.

Reviewer #2 (Remarks to the Author):

I find the revised manuscript to be significantly improved, and have only minor comments for additional changes. While the study may not "solve" the size control question in mammalian cells, it provides a wealth of solid data and interesting discussion that will aid the field in its long-term goals.

Answer: We thank the referee for his/her positive assessment of our manuscript.

Minor comments:

1. When discussing figures 4e-f on page 8, I would encourage the authors to avoid interpreting the "generally negative" slope. This trend is extremely weak, to the point of being largely inconclusive, at least for me. By interpreting these weak data, the authors lose ground on their interesting interpretation of stronger data throughout the manuscript.

Answer:

We agree with the referee that this result is unexpected and difficult to interpret. However, for both cell types, the bins appear to align along the slope predicted by the mechanistic adder (except for the last one for HT29). Thus, although we remain extremely cautious on its interpretation, we believe that it is important to mention this trend, which could motivate future investigations in this direction.

2. I have two suggested text changes for discussing the new data on primary cells on page 29 of the supplemental material. First, I would change "inhomogeneous" to "heterogeneous" on line 2. Second, the text on this page refers several times to "<Vbirth / Vmitosis>" when I believe they mean to indicate the reverse (i.e. <Vmitosis / Vbirth>).

Answer: We corrected the error and we thank the referee for spotting it.

Reviewer #4 (Remarks to the Author):

In this manuscript, Cadart et al. study the mechanisms of size control in several cell lines using FXm (fluorescence exclusion measurement). This method has the advantage of allowing the tracking of individual cells from birth to division in a controlled environment. Using this methodology, the authors describe how different cell types (and even subgroups of cells of the same cell type) can differently adapt the duration of G1 and S-G2 phase and their growth rate in order to achieve an effective near-adder behavior.

They conclude that while a phenomenological adder is commonly found in various cell types, there is not a universal underlying mechanism. These findings are integrated in a mathematical model that allows the comparison of size homeostasis behavior in different datasets. The authors had submitted a previous version of the manuscript for publication in Nature Cell Biology, and several concerns were raised. Since this first submission the authors have added new experimental data and have addressed the main criticisms posed by the reviewers. In general terms, I regard the quality of the experiments as high and I think that the data generated will be valuable to the field of size control. Hence, I recommend its publication in Nature Communications.

Answer: We are grateful to the referee for his/her positive assessment of our manuscript.

There are just a few points that I would like the authors to clarify/comment on:

1.- In the FXm experiments in supplementary figure 2, HT29-hgem cells contain a small sub-population with a longer G1 phase that add a larger volume over the cell cycle. In the figure the authors show the statistical analysis of the data once these outliers have been removed. However, it is unclear whether the same sub-population is also observed in subsequent experiments where this cell line is used (figures 3 and 4) (I assume it is), and, if so, whether they have been considered or excluded in the analysis. Do these cells with an abnormally long G1 phase modulate the length of S-G2?

Answer: As we explain in the legend of Supplementary Fig.2a, this subpopulation consists of the cells that have an added volume over the total cell cycle higher than $2000\mu\text{m}^3$. When we observed these outliers on the graph, we looked back at the raw data to see if there was no obvious reason explaining their presence: we checked that the points in this sub-population correspond to measurements coming from different replicates and that the fluorescence expression levels of hgeminin were comparable to those of the rest of the population. Since we could not find any biological argument to exclude this subpopulation, we kept it for all our analyses in Fig. 2, 3 and 4. The only consequence of this subpopulation is that it sometimes adds noise, but we always checked that it does not affect our conclusions. An illustration of such control is shown in Supplementary Fig. 2a for the analysis of the trends over the whole cell cycle, where the impact on the trend was most important. We did not show this control for modulation of time during S-G2 since in that case, this sub-population appears qualitatively similar to the rest of the population. Please see below, the plot of Δt_{S-G2} vs. $V_{G1/S}$, where the absence of modulation of S-G2 duration in HT29 is observed both in the main population and the sub-population.

Figure Legend: Plot of added volume in S-G2 as a function of the logarithm of volume at G1/S transition in HT29-hgem cells. This plot is the same as in Fig. 4a, except that the points are colored as follow: the points in orange correspond to the subpopulation showing an added volume over the whole cell cycle that is higher than $2000\mu\text{m}^3$, the points in blue correspond to cells where the added volume over the whole cell cycle was not measured, the points in green correspond to the rest of the point. Squares are median bins, errors bars are standard deviation.

2.- One of the points that the authors touch upon is the question of whether cell growth is linear or exponential. To address this matter, they analyze growth rate of HeLa cells (the slope of the plot of instantaneous growth speed as a function of volume). By grouping cells in three different categories according to their size they observe that, while in all cases there is a linear increase in growth speed as volume increases, the slope of the plot decreases for larger sizes at birth. Since the grouping of cells is arbitrary (20% smallest, 60% intermediate and 20% largest) I wonder whether a narrower grouping (in 5-10% steps) would show that the largest cells do not increase their growth speed as their volume increases, which would be indicative of linear growth after reaching a certain size.

Answer: This is an interesting point, unfortunately, with the current size of our dataset (although the trend we highlight is robust to binning) using narrower bins the noise becomes very high (particularly close to the tails of the distribution), making it impossible to address the question formulated by the referee.

3.- The advantage of the FXm method used throughout the manuscript is that cells grow without the space restriction of the microchannels. The authors argue that in the microchannels cells show indirect evidence of linear growth, whereas they observe exponential mode of growth with the FXm device. Nevertheless, in both cases size homeostasis seems to be achieved through an adder. Although cells are undoubtedly happier in the FXm setup than in the microchannels, it is unclear whether it reflects the natural mode of growth of cells in a tissue (where growth might be confined by neighboring cells). Have the authors considered analyzing cell growth in confluent areas of the FXm device or after seeding a higher density of cells?

Answer: This is a very interesting future research perspective for us, but we believe it goes beyond the frame of this study. Some of our collaborators are currently working on studying the growth of monolayers with FXm, while the study of single-cell growth in confluent monolayers would require further technological development (to date, the method only works on isolated cells since correct volume calculation relies on the integration of the fluorescence intensity signal over a larger area than that of the cell).